# DISP-LLM: Dimension-Independent Structural Pruning for Large Language Models

**Shangqian Gao** *
Florida State University

**Chi-Heng Lin**
Samsung Research America

**Ting Hua**
Samsung Research America

**Tang Zheng**
Samsung Research America

**Yilin Shen**
Samsung Research America

**Hongxia Jin**
Samsung Research America

**Yen-Chang Hsu**
Samsung Research America

## Abstract

Large Language Models (LLMs) have achieved remarkable success in various natural language processing tasks, including language modeling, understanding, and generation. However, the increased memory and computational costs associated with these models pose significant challenges for deployment on resource-limited devices. Structural pruning has emerged as a promising solution to reduce the costs of LLMs without requiring post-processing steps. Prior structural pruning methods either follow the dependence of structures at the cost of limiting flexibility, or introduce non-trivial additional parameters by incorporating different projection matrices. In this work, we propose a novel approach that relaxes the constraint imposed by regular structural pruning methods and eliminates the structural dependence along the embedding dimension. Our dimension-independent structural pruning method offers several benefits. Firstly, our method enables different blocks to utilize different subsets of the feature maps. Secondly, by removing structural dependence, we facilitate each block to possess varying widths along its input and output dimensions, thereby significantly enhancing the flexibility of structural pruning. We evaluate our method on various LLMs, including OPT, LLaMA, LLaMA-2, Phi-1.5, and Phi-2. Experimental results demonstrate that our approach outperforms other state-of-the-art methods, showing for the first time that structural pruning can achieve an accuracy similar to semi-structural pruning.

## 1 Introduction

Large Language Models (LLMs) have revolutionized the field of natural language processing by leveraging deep learning techniques to process and generate human-like text. Compared to smaller models, LLMs exhibit unique characteristics and demonstrate remarkable abilities in tackling a wide range of complex tasks [40]. Despite their impressive capabilities, the vast number of parameters in LLMs often hinders their deployment on resource-constrained devices, such as mobile phones. Consequently, there is significant interest in reducing the computational and memory requirements of LLMs.

Existing compression techniques for large language models (LLMs) include weight sparsification [9], structural pruning [30], and quantization [10]. In this work, we focus on structural pruning and

---

*Part of this project was completed at Samsung Research America. Correspondence to sgao@cs.fsu.edu

38th Conference on Neural Information Processing Systems (NeurIPS 2024).

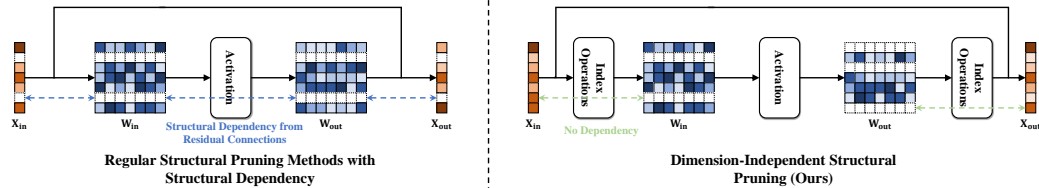

Figure 1: We use an MLP layer as an example. **Left:** Regular pruning methods have to follow structural dependence thus their flexibility is limited. **Right:** Our dimension-independent structural pruning method breaks the structural dependence via index operations and thus largely improves the flexibility for pruning.

address the limitations of previous methods in this category. Structural pruning [30] is a general-purpose compression solution that maintains LLM performance across various tasks, facilitates deployment on devices, and is computationally efficient. However, existing methods may restrict pruning flexibility or add significant overhead to the compressed model. For instance, LLM-Pruner [30] follows structural dependence during pruning, requiring different layers to use the same subset of feature maps, which limits pruning flexibility. SliceGPT [2] alleviates this issue by applying orthogonal projections for each layer but introduces a non-trivial number of additional parameters (e.g., **5% to 13% of the parameters** of the original model for LLaMA-2 7B). Our approach aims to overcome these drawbacks and offer a better performance-cost trade-off for structural pruning.

We aim to increase the flexibility of current structural pruning methods and consequently improve performance. Our method provides different sub-spaces or subsets of features to different layers, but unlike SliceGPT, it doesn't introduce additional parameters. To achieve this, we break the structural dependence of regular structural pruning methods, allowing different layers to have different subsets of features along the embedding dimension and an example is given in Fig. 1. After pruning, we employ index selection and index addition operations to sample subsets of features from the residual connection and add them back after the computation of each layer. Furthermore, our method introduces an additional level of flexibility by learning different widths for each layer. Our approach significantly improves the flexibility of structural pruning without adding additional parameters.

Extensive experimental results show that our method can outperform state-of-the-art structural pruning methods for LLMs while still maintaining low computational costs. Our method does not require recovery fine-tuning to obtain such performance. In addition, our method does not update the remained model weights during pruning which is a distinct departure from several other methods, such as SparseGPT [9] and LLM Surgeon [37]. Our contributions are as follows:

- We break the structural dependence of regular structural pruning methods, significantly increasing the flexibility of structural pruning. This allows different layers to select their own subset of features from the embedding dimension. Importantly, our method achieves this without introducing additional parameters, unlike SliceGPT.

- We propose to learn the widths of each layer using gradient-based optimization methods. A hypernetwork generates the column or row selection matrices, while the width of each layer is controlled globally. This approach allows for fine-grained control over the pruning process and enhances the adaptability of our method to various models and tasks.

- Our method demonstrates superior performance compared to state-of-the-art structural pruning techniques for LLMs across a range of models, including OPT, LLaMA, LLaMA-2, Phi-1.5, and Phi-2. Notably, the resulting model from our method is a sub-network that exists within the original model, indicating the effectiveness of our method in discovering strong sub-networks.

## 2   Related Works

Magnitude-based pruning is the most straightforward approach to reduce model size, where weights with the smallest magnitude are pruned. *Han et al.* [14] employ this strategy for pruning with $L_1$ or $L_2$ norm of weights. Filter pruning [24] extends this setting to structures of the model instead of performing weight-level sparsification. Although magnitude-based pruning methods are very efficient, they result in significant performance drops for LLMs, even for weight pruning [9]. Another

line of research, Optimal Brain Damage [23] and Optimal Brain Surgeon [15], utilize second-order information to remove connections. These methods require calculating the inverse of the Hessian matrix, which is computationally intensive for modern neural network architectures like Convolutional Neural Networks (CNNs) [22, 16], Transformers [38], or Large Language Models (LLMs) [35]. To reduce the cost of computing the Hessian inverse matrix, Optimal Brain Surgeon can be applied in a layer-wise fashion [7, 8], making the computation tractable. However, further scaling up these methods for LLMs remains challenging.

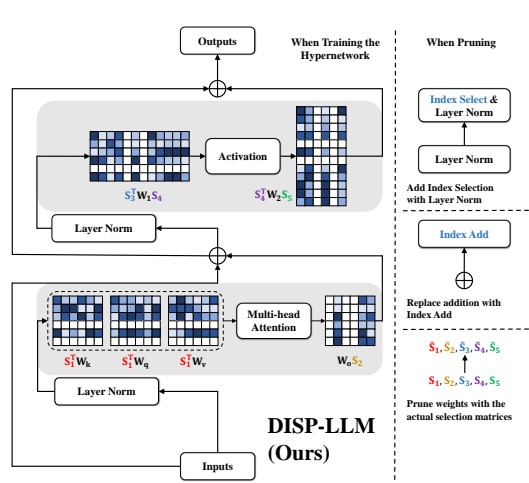

Figure 2: Our method, DISP-LLM, applies different selection matrices to the input and output dimension of the Attention layer and MLP layer ($\mathbf{S}_1/\mathbf{S}_2$: Attention in/out; $\mathbf{S}_3/\mathbf{S}_4/\mathbf{S}_5$: MLP in/middle/out). When pruning the model, we add "Index Selection" before Layer Norm and we replace addition with "Index Add." $\hat{\mathbf{S}}_1, \cdots, \hat{\mathbf{S}}_5$ are applied for pruning weight matrices.

Recent methods like SparseGPT [9] or GPTQ [10] aim to minimize the squared error before and after pruning or quantization of a given layer. In this setting, the Hessian inverse matrix becomes easy to compute, as it is simply the multiplication between the feature map and its transpose for a given layer. GPTQ and SparseGPT then quantize or sparsify model weights in a column-by-column manner, and the unpruned or unquantized weights are updated to compensate for the error of pruning and quantization. Wanda [34] further avoids computing the inverse of the Hessian matrix by only considering the diagonal of the Hessian matrix. While SparseGPT and Wanda achieve good results, unstructured sparsity is known to be harder to achieve actual speedup. They also applied their methods on semi-structured settings [31], but the performance becomes much worse.

Several researches [28, 19, 44, 13, 42, 12] apply learnable parameters for specific structures when pruning vision or language models. However, many of these methods cannot be scaled up to LLMs since they need to learn weights and structures together. In contrast, our method explores sub-networks within the original model without updating model weights. Additionally, our method mainly explores the regime of pruning without recovery fine-tuning, which is rarely presented in previous methods with learnable parameters on structures. Our method is also related to the unconstrained channel pruning for CNNs [39]. However, our method explores this idea from the perspective of breaking structural dependence and scales it to much larger models than [39]. Moreover, our method thoroughly explores the global allocation of parameters, where [39] fails to do.

Recently, several works have been proposed to reduce the size of LLMs. LLM-Pruner [30] aims to remove connected structures using importance calculated from Taylor expansions. SliceGPT [2] offers more flexibility than regular pruning by projecting the feature maps to different spaces but introduces extra parameters in the residuals. LLM Surgeon [37] periodically updates model weights and structures, resulting in a higher cost than LLM-Pruner and SliceGPT. Our proposed DISP-LLM breaks the structural dependence relied on by LLM-Pruner, without additional transformation matrices in the residual connections like SliceGPT. Furthermore, in contrast to LLM Surgeon, which requires extensive computational resources, our method is significantly more efficient.

## 3 Preliminary

### 3.1 Notations

To better understand our paper, we first define some notations. We use $d$ to denote the model dimension or embedding dimension of LLMs. $\mathbf{X} \in \Re^{b \times n \times d}$ is used to represent feature maps, and $b$ is the mini-batch size, $n$ is the number of tokens. $\mathbf{W} \in \Re^{d_1 \times d_2}$ is the model weights of size $d_1 \times d_2$. Let $\mathbf{S}$ denote a pseudo-index selection matrix of size $d \times d$, which is a diagonal matrix filled with 0 or 1 and the positions of the ones indicate the selected index. We further use $\hat{\mathbf{S}}$ of size $d \times d_{\text{small}}$ to

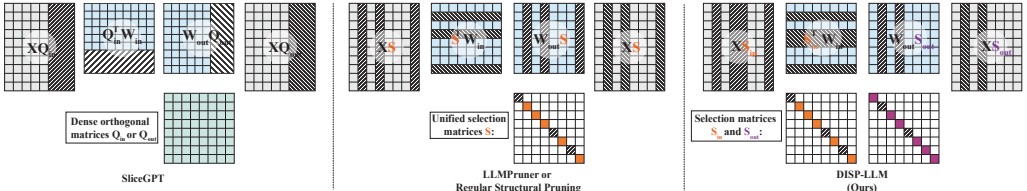

Figure 3: Comparison of the projection matrices for structural pruning. We use $\mathbf{W}_{\text{in}}$ and $\mathbf{W}_{\text{out}}$ in Fig. 1 as an example. **Left:** SliceGPT employs orthogonal projection matrices, and it has to insert the projection matrices into the residual connections. **Middle:** Regular structural pruning methods remove structures based on their dependence, requiring to use the unified selection matrix $\mathbf{S}$ for all blocks, which limits flexibility. **Right:** Our method breaks the structural dependence, allowing the use of different selection matrices $\mathbf{S}_{in}$ and $\mathbf{S}_{out}$ for the embedding dimension, significantly improving the flexibility of pruning.

represent the actual selection matrix by removing $d - d_{\text{small}}$ columns with all zeros from $\mathbf{S}$. For any matrix $\mathbf{A}$, nnz($\mathbf{A}$) represents the number of nonzero entries of $\mathbf{A}$.

### 3.2 Revisit SliceGPT

The core idea of SliceGPT [2] is to achieve computational invariance within the transformer architecture. It demonstrates that orthogonal projections can be applied to the output of each block and subsequently undone in the next block. This transformation is computed using Principal Component Analysis (PCA), allowing the feature maps between blocks to be projected into their principal components. A significant advantage of this approach is that it projects the feature maps of different blocks into distinct spaces, thereby introducing an additional degree of freedom for compression. This flexibility is not captured by regular structural pruning methods like LLM-Pruner [30], which rely on structural dependence.

After slicing (pruning), the feature map and weight matrix of $l$th layer of SliceGPT become:

$$\tilde{\mathbf{X}}_l = \mathbf{X}_l \mathbf{Q}_l \hat{\mathbf{S}}, \ \tilde{\mathbf{W}}_l = \hat{\mathbf{S}}^\top \mathbf{Q}_l^\top \mathbf{W}_l. \tag{1}$$

where $\hat{\mathbf{S}}$ is a $d \times d_{\text{small}}$ selection matrix, $\mathbf{X}_l$ is the output of the $l$th block, and $\mathbf{Q}_l$ contains eigenvectors of $\mathbf{C}_l$:

$$\mathbf{C}_l = \sum_i \mathbf{X}_{l,i}^\top \mathbf{X}_{l,i}$$

and $\mathbf{X}_{l,i}$ is the $i$-th column of $\mathbf{X}_l$ (corresponding to the $i$th sequence in the calibration dataset). From Eq. 1, we can see that SliceGPT uses the same selection matrix $\hat{\mathbf{S}}$ for all layers, but the feature map $\mathbf{X}_l$ is firstly projected by $\mathbf{Q}_l$, and the pruning for different layers is along with different directions. One crucial drawback of SliceGPT also comes from the projection matrix $\mathbf{Q}_l$, since the residual connection must be multiplied by the linear transformation $\mathbf{Q}_l^\top \mathbf{Q}_{l+1}$ (shown in Fig. 7 **left** in the Appendix). These additional operations bring a non-trivial amount of additional parameters. For a model that has $L$ blocks, with the model dimension $d$ and the remaining percentage of parameters $p \in [0, 1]$, it brings approximately $Ld^2p^2$ additional parameters to the model (more than **10% of model parameters** in some cases, and more details are given in Fig 10 in the Appendix).

### 3.3 Residual Connections Limit the Flexibility of Structural Pruning

SliceGPT offers significant flexibility, but achieving similar flexibility with regular structural pruning methods without adding extra parameters is challenging. This section explains the reasons behind this difficulty. To simplify our reasoning, we replace $\hat{\mathbf{S}}$ with its pseudo selection matrix $\mathbf{S}$.

Assume we follow the basic setting of dependence-based structural pruning but allow each layer the flexibility to have its own selection matrix, $\mathbf{S}_l$, along the embedding dimension. Under this assumption, due to structural dependence, all layers will share the same width of nnz($\mathbf{S}_0 \mathbf{S}_1 \cdots \mathbf{S}_L$).

In order to prune different positions for different layers, we need to add a transformation matrix to align the width of layers $l$ and $l + 1$. Intuitively, if we have $\mathbf{S}_l$ and $\mathbf{S}_{l+1}$, we can then insert $\mathbf{S}_l^\top \mathbf{S}_{l+1}$ in the residual connection to align consecutive layers.

**Algorithm 1:** Block inference after pruning.

---

**Input**: Feature map of the previous block $\mathbf{X}_{\text{in}}$. Preserved indices sets $\mathbf{Ind}_1, \mathbf{Ind}_2, \mathbf{Ind}_3, \mathbf{Ind}_5$.

1. $\hat{\mathbf{X}}_{\text{in}} = \text{LayerNorm}(\mathbf{X}_{\text{in}}[:, \mathbf{Ind}_1])$.                    ▷ *Index Selection for Attention*

2. $\mathbf{X}_{\text{att}} = \text{MultiHead}(\hat{\mathbf{X}}_{\text{in}}\hat{\mathbf{S}}_1^\top \mathbf{W}_q, \hat{\mathbf{X}}_{\text{in}}\hat{\mathbf{S}}_1^\top \mathbf{W}_k, \hat{\mathbf{X}}_{\text{in}}\hat{\mathbf{S}}_1^\top \mathbf{W}_v)\mathbf{W}_o\hat{\mathbf{S}}_2$.

3. $\mathbf{X}_{\text{res}} = \text{Index\_Add}(\mathbf{X}_{\text{in}}, \mathbf{X}_{\text{attn}}, \mathbf{Ind}_2)$.                    ▷ *Index Addition with the input*

4. $\hat{\mathbf{X}}_{\text{res}} = \text{LayerNorm}(\mathbf{X}_{\text{res}}[:, \mathbf{Ind}_3])$.                    ▷ *Index selection for MLP*

5. $\mathbf{X}_{\text{mlp}} = (\sigma(\hat{\mathbf{X}}_{\text{res}}\hat{\mathbf{S}}_3^\top \mathbf{W}_1\hat{\mathbf{S}}_4) \odot (\hat{\mathbf{X}}_{\text{res}}\hat{\mathbf{S}}_3^\top \mathbf{W}_2\hat{\mathbf{S}}_4))\hat{\mathbf{S}}_4^\top \mathbf{W}_3\hat{\mathbf{S}}_5$.

6. $\mathbf{X}_{\text{out}} = \text{Index\_Add}(\mathbf{X}_{\text{res}}, \mathbf{X}_{\text{mlp}}, \mathbf{Ind}_5)$                    ▷ *Index Addition with the residual*

**Return** $\mathbf{X}_{\text{out}}$ for the next block.

---

With this setup, we can use $\mathbf{X}_l\mathbf{S}_l$ to select subsets of features for different layers, mimicking $\mathbf{Q}_l\mathbf{S}$ for SliceGPT. Although it seems promising, this formulation has issues with layer widths, as detailed in Proposition 1.

**Proposition 1** (Decreasing feature dimensions for deeper layers). *Let the pseudo-selection matrices in layers $l$ and $l + 1$ be $\mathbf{S}_l$ and $\mathbf{S}_{l+1}$, respectively. The number of nonzero entries in the residual adapter satisfies*

$$nnz(\mathbf{S}_l^\top \mathbf{S}_{l+1}) \leqslant \min\{nnz(\mathbf{S}_l), nnz(\mathbf{S}_{l+1})\}.$$

*For compression strategies that remove dependent structures for layer $l + 1$ following $\mathbf{S}_l^\top \mathbf{S}_{l+1}$, this implies that the dimension in layer $l + 1$ is less than or equal to that in layer $l$, with equality holding when the feature indices selected in layer $l + 1$ are contained within those in layer $l$ or vice versa.*

**Remark.** The proof of Proposition. 1 is straightforward and it is given in the Appendix A.1. From Proposition 1, we observe that if we naively apply $\mathbf{S}_l$ for different layers, the model width will progressively decrease as we go deeper into the network. It also fails to provide different sets of features for different layers; instead, it merely passes a subset of features from the previous layer to the next. To avoid this restriction, all blocks must share the same width and the same pruned columns or rows. And we then fall back to the regime of previous structural pruning methods such as LLM-Pruner [30], Shared LLaMA [43], etc.

Proposition 1 highlights two significant obstacles. First, dependence-based structural pruning methods result in a uniform width along the embedding dimension. Second, inserting selection matrices in the residual connections causes the embedding dimension to decrease with depth. These challenges are unavoidable due to the residual connections linking structures across layers. To enhance flexibility along the embedding dimension, bypassing the residual connections is crucial.

## 4 Dimension-Independent Large Language Model

### 4.1 Break the Structural dependence

Section 3.3 demonstrates that the residual connection is the primary barrier preventing pruning methods from achieving better flexibility. To avoid modifying the residual connection, we relocate the selection matrices inside the residual connection. This approach allows us to successfully create different subsets from the feature maps for different layers.

Based on this idea, we propose a solution that involves pruning different positions in consecutive blocks and selecting or merging feature maps from or back to the residual connection. This approach breaks the structural dependence inherent in previous pruning methods. Formally, given a transformer block, we apply the following operations:

$$\text{Attention}(\mathbf{X}) = \text{MultiHead}(\mathbf{X}\mathbf{S}_1^\top \mathbf{W}_q, \mathbf{X}\mathbf{S}_1^\top \mathbf{W}_k, \mathbf{X}\mathbf{S}_1^\top \mathbf{W}_v)\mathbf{W}_o\mathbf{S}_2, \tag{2}$$

$$\text{MLP}(\mathbf{X}) = (\sigma(\mathbf{X}\mathbf{S}_3^\top \mathbf{W}_1\mathbf{S}_4) \odot (\mathbf{X}\mathbf{S}_3^\top \mathbf{W}_2\mathbf{S}_4))\mathbf{S}_4^\top \mathbf{W}_3\mathbf{S}_5, \tag{3}$$

where $\mathbf{S}_1, \ldots, \mathbf{S}_5$ are pseudo selection matrices of size $d \times d$, and $\odot$ denotes element-wise multiplication. Eq. 3 gives an example operation for gated MLP modules used in LLaMA [35]. For Phi models [1] or OPT [46], the MLP operation is defined as $\text{MLP}(\mathbf{X}) = \sigma(\mathbf{X}\mathbf{S}_3^\top \mathbf{W}_1\mathbf{S}_4)\mathbf{S}_4^\top \mathbf{W}_3\mathbf{S}_5$. Fig 2 illustrates how to insert these selection matrices into a transformer block.

Given the operations defined in Eq. 2 and Eq. 3, we successfully remove the constraint in Proposition 1. The input and output of both the Attention layer and the MLP layer can be selected differently from the original feature maps for different layers, mimicking the function of $\mathbf{Q}_l$ in SliceGPT. Additionally, our method eliminates the need for extra parameters in $\mathbf{Q}_l^\top \mathbf{Q}_{l+1}$ as it does not alter the residual connection. We also enhance flexibility by pruning the middle dimension of the MLP layer. Additionally, this flexibility can be further improved by allowing the query, key, and value weight matrices to use different selection matrices. Our current form is kept for two reasons: (1) SliceGPT uses one $\mathbf{Q}_l$ per layer, and we followed this design for a fair comparison, and (2) adding separate selection matrices would increase indexing operations, potentially slowing down the inference. Fig 3 further compares the projection matrices for SliceGPT, regular structural pruning, and the proposed method.

Once we have the final selection matrices $\mathbf{S}_1, \ldots, \mathbf{S}_5$, the pruned model will use a combination of index selection and index addition for inference as shown in Algorithm 1, where $\mathbf{Ind}_i$ is a set containing all indices equal to one in the diagonal of $\mathbf{S}_i$:

$$\mathbf{Ind}_i = \{j \mid \text{if } \mathbf{s}_{i[j]} = 1\}, \ \mathbf{s}_i = \text{diag}(\mathbf{S}_i).$$

The same color is used to mark the index set $\mathbf{Ind}_i$ and its corresponding selection matrix $\hat{\mathbf{S}}_i$. Index_Add($\mathbf{A}, \mathbf{B}, \mathbf{Ind}$) adds matrices $\mathbf{A}$ and $\mathbf{B}$ along the last dimension on selected positions from $\mathbf{Ind}$, then returns $\mathbf{A}$ after index addition. With index selection and addition, the block dimension can be freely changed. Index selection and addition introduce some overhead, but as demonstrated in the experiment section, we still observe improvements in throughput.

## 4.2 Learning the Width for Dimension-Independent LLMs

Building on the dimension-independent setting introduced in Section 4.1, our approach offers much greater flexibility in selecting sub-networks from the original dense model compared to the constrained settings in LLM-Pruner [30]. The next challenge is determining the width of each layer. Given the large search space of our dimension-independent structural pruning and the computationally intensive nature of LLMs, it is impractical to use reinforcement learning [17] or evolutionary search-based algorithms [27]. Therefore, we adopt gradient-based methods to address this challenge. Given the diagonal vector $\mathbf{s}_i \in \{0,1\}^d$ from $\hat{\mathbf{S}}_i$, the Straight-Through (ST) gradient estimator [3] is used to estimate the gradients with respect to learnable continuous latent parameters. More specifically, we use the recently proposed gradient estimator ReinMax [26] to estimate the gradients through the binary operation. A detailed explanation of ReinMax for the binary case is provided in Appendix A.2.

Given the large search space of our method, we find that only using element-wise learnable parameters is insufficient. To address this issue, a hypernetwork is introduced to generate latent parameters for ReinMax, as detailed below:

$$\mathbf{s} = \text{ReinMax}(\text{HyperNetwork}(\Theta)), \tag{4}$$

where $\Theta$ represents the parameters of the hypernetwork and $\mathbf{s}$ contains $\mathbf{s}_i$ from all blocks. The hypernetwork is composed of GRU [5] and fully connected layers, where the GRU captures block-wise relationships and the fully connected layers capture relationships between different dimensions. With the hypernetwork and ReinMax, we can effectively learn the width of each block. The details of the hypernetwork are provided in Appendix A.3.

## 4.3 Dimension-Independent Structural Pruning as an Optimization Problem

With the methods described above, we can formulate dimension-independent structural pruning as an optimization problem, with regularization to control the number of remaining parameters. We insert $\mathbf{s}$ back into $\mathbf{S}$ as defined in section 4.1 for forward and backward calculations. The overall objective function is listed below:

$$\min_{\Theta} \ \mathcal{L}(\mathcal{X}; \mathbf{W}, \mathbf{s}) + \lambda \mathcal{R}(T(\mathbf{s}), pT_{\text{total}}), \tag{5}$$

$$\mathcal{R}(T(\mathbf{s}), pT_{\text{total}}) = \log(\max(T(\mathbf{s}), pT_{\text{total}}) / \min(T(\mathbf{s}), pT_{\text{total}})), \tag{6}$$

where $\mathcal{L}$ is the language modeling loss function of next word prediction, $\mathcal{X}$ represents the input tokens, $\mathbf{W}$ is the collection of model weights, $\mathbf{s}$ is defined in Eq. 4, and $\mathcal{R}$ is a parameter regularization loss function defined in Eq. 6. Here, $T(\mathbf{s})$ denotes the number of parameters controlled by the current structure $\mathbf{s}$, $T_{\text{total}}$ is the total number of parameters of the model, and $p \in (0, 1]$ is a user-defined

Table 1: Perplexities of different structural pruning methods on WikiText-2. Our method is the only one that does not update model weights. SliceGPT does not directly update model weights, however, it applies orthogonal transformation matrices to the weights.

| Method | Pruning Ratio | W Update? | Test Performance (PPL) | | | | | |
|---|---|---|---|---|---|---|---|---|
| | | | OPT 125M | OPT 1.3B | OPT 2.7B | OPT 6.7B | LLaMA-2 7B | LLaMA-2 13B |
| Dense | 0% | - | 27.65 | 14.62 | 12.47 | 10.86 | 5.12 | 4.57 |
| SliceGPT [2] | 10% | ✗ | 29.34 | 15.10 | 12.75 | 10.92 | 5.89 | 5.21 |
| | 20% | ✗ | 34.26 | 16.43 | 13.73 | 11.48 | 6.64 | 5.81 |
| | 25% | ✗ | 37.74 | 17.46 | 14.56 | 11.90 | 7.24 | 6.30 |
| | 30% | ✗ | 43.98 | 19.09 | 15.83 | 12.51 | 8.12 | 6.99 |
| K-OBD [34] | 20% | ✓ | 29.89 | 15.63 | 12.47 | 11.28 | 9.14 | 6.29 |
| | 30% | ✓ | 36.54 | 18.29 | 14.53 | 13.03 | 15.43 | 10.08 |
| | 40% | ✓ | 47.54 | 24.65 | 18.09 | 16.21 | 28.03 | 13.06 |
| | 50% | ✓ | 75.95 | 37.68 | 26.68 | 25.54 | 46.64 | 16.06 |
| LLM Surgeon [34] | 20% | ✓ | 28.73 | 15.12 | 12.27 | 11.02 | 6.18 | 5.29 |
| | 30% | ✓ | 31.82 | 16.24 | 12.92 | 11.64 | 7.83 | 6.21 |
| | 40% | ✓ | 38.47 | 18.45 | 14.23 | 12.58 | 10.39 | 7.25 |
| | 50% | ✓ | 49.78 | 22.95 | 17.15 | 14.90 | 15.38 | 9.43 |
| DISP-LLM (Ours) | 20% | ✗ | 25.21 | 13.12 | 11.72 | 9.89 | 6.10 | 5.21 |
| | 30% | ✗ | 28.16 | 14.79 | 12.16 | 10.90 | 6.85 | 5.77 |
| | 40% | ✗ | 34.31 | 17.77 | 14.11 | 12.18 | 8.11 | 6.59 |
| | 50% | ✗ | 39.87 | 21.70 | 17.07 | 14.06 | 9.84 | 7.11 |

Table 2: Comparison of our method against semi-structure pruning methods on WikiText-2.

| Method | Pruning Ratio | W Update? | Structure? | Test Performance (PPL) | | | |
|---|---|---|---|---|---|---|---|
| | | | | LLaMA 7B | LLaMA 13B | LLaMA-2 7B | LLaMA-2 13B |
| Dense | 0% | - | - | 5.68 | 5.09 | 5.12 | 4.57 |
| Magnitude | 2:4 | ✗ | ✗ | 42.13 | 18.37 | 54.59 | 8.33 |
| SparseGPT [9] | 2:4 | ✓ | ✗ | **11.00** | 9.11 | 10.17 | 8.32 |
| Wanda [34] | 2:4 | ✗ | ✗ | 11.53 | 9.58 | 11.02 | 8.27 |
| DISP-LLM (ours) | 50% | ✗ | ✓ | 11.47 | **8.15** | **9.84** | **7.11** |

parameter to control how many parameters should be preserved within the model. With the objective function in Eq. 5, the structures for dimension-independent pruning can be efficiently optimized. Moreover, the overhead of our method is minimal and comparable to parameter-efficient fine-tuning methods like LoRA [18], as it does not require storing gradients or the first and second-order momentum of model weights for the Adam optimizer [21].

# 5 Experiments

## 5.1 Settings

**Models.** We evaluate our **DISP-LLM** method using several LLMs with transformer blocks. Specifically, we choose the following models: OPT [46]: OPT-125M, OPT-1.3B, OPT-2.7B, OPT-6.7B; Phi-1.5 [25] and Phi-2 [20]; LLaMA 7B [35]; LLaMA-2 [36]: LLaMA-2 7B and LLaMA-2 13B.

**Implementations.** We implemented our method using Pytorch [32] and Hugging Face transformer library [41]. We freeze the model weights **W** when training the hypernetwork. We use the AdamW [29] optimizer to optimize the hypernetwork. The hypernetwork is trained for 10,000 iterations for all models. For all experiments, we set $\lambda$ in Obj. 5 to 6. Depending on the size of the base model, we use 1 to 4 NVIDIA A100 GPUs to train the hypernetwork. More implementation details can be found in the Appendix A.4.

**Datasets.** Following previous papers [2, 30], we use WikiText-2 and Alpaca datasets to train the hypernetwork. Following SliceGPT [2], we evaluate our method and other methods on five well-known zero-shot tasks: PIQA [4]; WinoGrande [33]; HellaSwag [45]; ARC-e and ARC-c [6]. We use llm-eval-harness [11] to evaluate the compressed models.

**Baselines.** We compare our **DISP-LLM** across baselines from structural pruning like LLM-Pruner [30], SliceGPT [2] and LLMSurgeon [37]. We also include semi-structure pruning baselines like SparseGPT [9] and Wanda [34].

## 5.2 Language Modeling

In Table 1, we report the perplexity of pruned OPT and LLaMA-2 models. Our DISP-LLM, which does not update weights, consistently outperforms more complex pruning methods such as K-OBD and LLM Surgeon, which involve weight updates, across **all pruning ratios and models**. The

Table 3: Zero-shot performance of the compressed LLaMA 7B, LLaMA-2 7B and Phi models. The structure of *DISP-LLM* is based on the WikiText dataset, and the structure of *DISP-LLM Alpaca* is based on the Alpaca dataset.

| Pruning Ratio | Method | W Update? | WinoGrande acc | HellaSwag acc-norm | ARC-e acc-norm | ARC-c acc-norm | PIQA acc-norm | Avg |
|---|---|---|---|---|---|---|---|---|
| 0% | LLaMA 7B | - | 69.85 | 76.21 | 72.81 | 44.71 | 79.16 | 68.55 |
| 20% | LLM-Pruner [30] | ✗ | 61.33 | 65.34 | 59.18 | 37.12 | 75.57 | 59.71 |
| | +finetuning | ✓ | 65.11 | 68.11 | 63.43 | **37.88** | 76.44 | 62.19 |
| | DISP-LLM (Ours) | ✗ | **66.54** | **68.75** | 59.60 | 35.24 | 74.97 | 61.02 |
| | DISP-LLM Alpaca (Ours) | ✗ | 64.72 | 68.39 | **64.81** | 37.12 | **76.66** | **62.34** |
| 50% | LLM-Pruner [30] | ✗ | 53.20 | 35.64 | 33.50 | 27.22 | 59.63 | 41.84 |
| | +finetuning | ✓ | 55.09 | 47.56 | 46.46 | 28.24 | **68.82** | 49.23 |
| | DISP-LLM (Ours) | ✗ | **58.41** | 47.71 | 44.40 | 28.50 | 64.09 | 48.62 |
| | DISP-LLM Alpaca (Ours) | ✗ | 56.91 | **48.76** | **48.91** | **31.57** | 67.46 | **50.72** |
| 0% | LLaMA-2 7B | - | 69.14 | 75.99 | 74.58 | 46.15 | 79.11 | 68.99 |
| 30% | SliceGPT [2] | ✓ | 61.33 | 49.62 | 51.77 | 31.23 | 63.55 | 51.50 |
| | K-OBD [34] | ✓ | 56.83 | 53.07 | 51.05 | 33.11 | 71.82 | 53.18 |
| | LLM Surgeon [34] | ✓ | 61.09 | 60.72 | **63.09** | 36.69 | 73.56 | 59.03 |
| | DISP-LLM (Ours) | ✗ | 62.27 | **63.43** | 59.81 | 33.19 | 71.82 | 58.10 |
| | DISP-LLM Alpaca (Ours) | ✗ | **63.93** | 62.87 | 60.10 | **37.03** | **73.72** | **59.53** |
| 50% | K-OBD [34] | ✓ | 53.04 | 36.84 | 36.11 | 26.71 | 60.66 | 42.67 |
| | LLM Surgeon [34] | ✓ | 52.57 | 40.29 | 44.91 | 26.28 | 64.36 | 45.68 |
| | DISP-LLM (Ours) | ✗ | 54.54 | 46.33 | 43.06 | 25.85 | 63.93 | 46.72 |
| | DISP-LLM Alpaca (Ours) | ✗ | **56.20** | **49.35** | **51.14** | **30.20** | **68.34** | **51.05** |
| 0% | Phi-1.5 | - | 72.77 | 62.58 | 73.11 | 48.04 | 75.63 | 66.43 |
| 30% | SliceGPT [2] | ✓ | **64.96** | 42.54 | 53.66 | 31.91 | 65.45 | 51.52 |
| | DISP-LLM (Ours) | ✗ | 61.48 | **47.97** | **57.66** | **33.01** | **71.08** | **54.24** |
| 0% | Phi-2 | - | 75.61 | 73.86 | 78.24 | 54.01 | 79.11 | 72.17 |
| 30% | SliceGPT [2] | ✓ | 63.14 | 47.56 | 53.03 | 30.29 | 65.94 | 51.99 |
| | DISP-LLM (Ours) | ✗ | **65.19** | **54.43** | **63.59** | **38.48** | **73.34** | **59.00** |

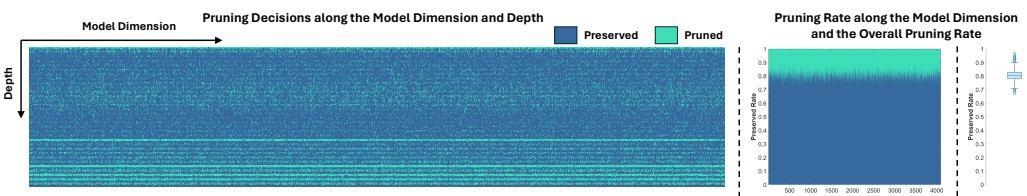

Figure 4: The pruned model architecture along the embedding dimension (model dimension) for the LLaMA-2 7B model when the pruning ratio equals 50%.

performance gap is even larger when compared to SliceGPT. The advantage is particularly clear in better-trained models like LLaMA-2 7B and 13B. For instance, our method surpasses LLM Surgeon by margins of 5.54 and 2.22 when pruning 50% of parameters of LLaMA-2 7B and 13B, respectively. Against K-OBD, our performance advantage extends to 36.80 and 9.49 under the same setting. For consistency, we let the pruning ratio of SliceGPT equal the slicing ratio. However, the actual pruning ratio for SliceGPT is much lower than the slicing ratio. More details are given in Appendix A.5.

In Table 2, we report the perplexity of pruned LLaMA and LLaMA-2 models and we compare our method with semi-structure pruning methods. From the table, we can see that our method outperforms both SparseGPT and Wanda on LLaMA 13B and LLaMA-2 7B/13B models. Our method performs on par with SparseGPT and Wanda with the LLaMA 7B model, and our DISP-LLM is a little bit worse than SparseGPT and is similar to Wanda. We are the first to show that **structural pruning methods can have a better or similar performance than semi-structural pruning methods**.

### 5.3 Zero-shot Performance

In Tab. 3, we present the zero-shot performance of the pruned model. For the LLaMA 7B model, we compare our method against LLM-Pruner with and without recovery fine-tuning. Our method consistently outperforms LLM-Pruner without fine-tuning, and the gap ranges from 2.63 to 8.88 across different pruning rates for average task performance. After fine-tuning, the performance of LLM-Pruner is largely boosted, however, our method is still able to outperform it demonstrating the existence of strong sub-networks within the original model. For the LLaMA-2 7B model, we compare our method against SliceGPT, K-OBD, and LLM Surgeon. With weight updates, LLM

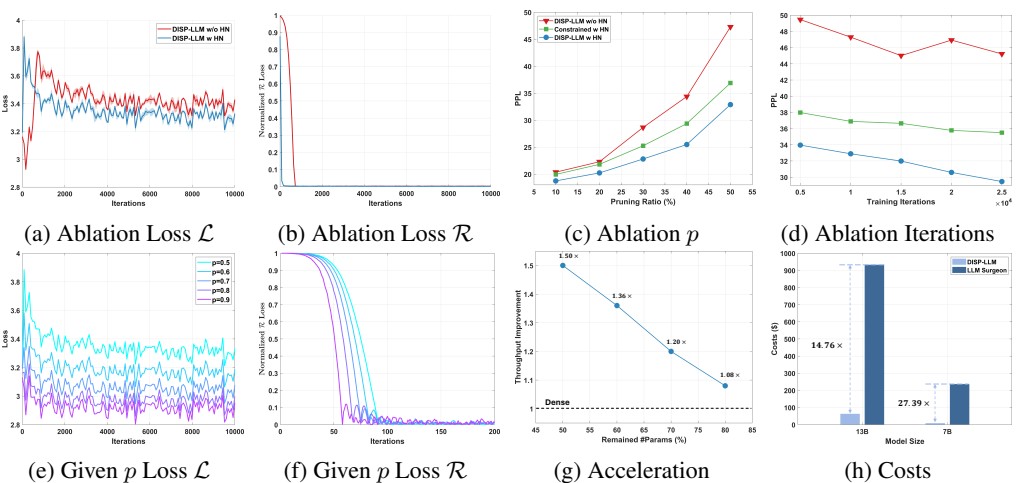

| (a) Ablation Loss $\mathcal{L}$ | (b) Ablation Loss $\mathcal{R}$ | (c) Ablation $p$ | (d) Ablation Iterations |
|---|---|---|---|
| (e) Given $p$ Loss $\mathcal{L}$ | (f) Given $p$ Loss $\mathcal{R}$ | (g) Acceleration | (h) Costs |

Figure 5: The training dynamics when learning the hypernetwork are shown in Figs. 5a, 5b, 5e, 5f. The results of different settings are in Figs. 5c, 5d, throughput is in Fig. 5g, and cost is in Fig. 5h.

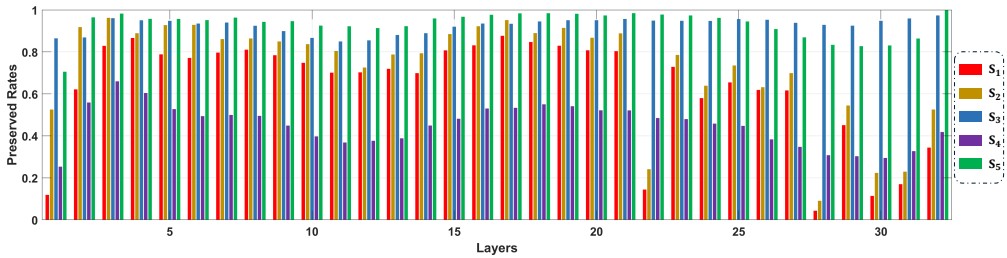

Figure 6: Model width after pruning for the LLaMA-2 7B model when the pruning ratio equals 50%.

Surgeon performs well with a lower pruning ratio like 30%. At this pruning ratio, our method performs similarly to LLM Surgeon, and our method outperforms other comparison baselines. When increasing the pruning ratio to 50%, the advantage of our method becomes obvious, and the gap between our method and LLM Surgeon is 5.37 for average task performance. We further compare our method with SliceGPT on Phi-1.5 and Phi-2, and our method consistently achieves better performance.

### 5.4 Analysis

**Ablation Study.** We visualize the results of ablation studies in Figs. 5a, 5b, 5c, 5d with Phi-1.5 model. The setting *"DISP-LLM w/o HN"* refers to using element-wise gates for learning sub-networks. The setting *"Constrained LLM w HN"* refers to pruning the embedding dimension following the structural dependence like LLM-Pruner. From Figs. 5a, 5b, we can see that using the hypernetwork largely accelerates the learning process for DISP-LLM, which is also verified in Figs. 5c, 5d. From Figs. 5c, 5d, we also see that our DISP-LLM always outperforms constrained structural pruning, demonstrating the value of added flexibility by breaking the dependence. To further study the impact of the HyperNetwork architecture, we provide more results in Tab. 4. *"w/o HN"* is equivalent to *"DISP-LLM w/o HN"*. The setting *"w/o Bi-GRU"* simply removes GRU and adds a fixed input (initialized in the same way as see Appendix A.3 for more details) for each linear layer. These results indicate that both GRU and linear layers within the HyperNetwork affect the final performance. One explanation is that linear layers connect different dimensions of the model, accelerating learning, while GRU layers capture inter-layer relationships, further reducing the difficulty of learning sub-network structures. Therefore, both GRU and linear layers are essential to the HyperNetwork.

**Different Pruning Ratios, Costs and Throughput.** In Figs. 5e, 5f, we show the language modeling loss $\mathcal{L}$ and regularization loss $\mathcal{R}$ in Obj 5 given different pruning ratios $p$ with Phi-1.5 model. We can see that our method consistently minimizes the language modeling loss across different $p$. In addition,

Table 4: The impact of the Hypernetwork architecture on the Phi-1.5 model. Performance is measured by PPL (perplexity).

| Settings | Compression Rate | | | | | |
|---|---|---|---|---|---|---|
| | 0% | 10% | 20% | 30% | 40% | 50% |
| w/o HN | | 20.37 | 22.30 | 28.66 | 34.33 | 47.29 |
| w/o Bi-GRU | 21.82 | 19.90 | 21.65 | 26.11 | 30.88 | 37.43 |
| Full HyperNetwork | | 18.75 | 20.23 | 22.81 | 25.49 | 32.89 |

our method quickly pushes the regularization loss $\mathcal{R}$ to near 0 values within 200 iterations. In Fig. 5g, the pruned model from LLaMA-13B improves the throughput of the dense model by $1.08\times$ to $1.50\times$. In Fig. 5h, we compare the costs of our method against LLM Surgeon. Our method is $\mathbf{27.39\times}$ and $\mathbf{14.76\times}$ cheaper compared to LLM Surgeon with LLaMA-2 7B and LLaMA-2 13B models.

**Every Embedding Dimension is Important.** In Fig. 4, we visualize the pruning decisions along the embedding dimension and depth for the LLaMA-2 7B model, we can see that all embedding dimensions have been used across different layers. This becomes more obvious in the second right figure of Fig. 4, where we sum all pruning decisions along the depth dimension, and we can see that every embedding dimension is kept around $80\%$ given all layers. We further visualize the model width after pruning for the LLaMA-2 7B model in Fig. 6, where we can see that several layers are more severely pruned than other layers.

# 6 Conclusion

In this paper, we proposed dimension-independent structural pruning for Large Language Models. By breaking the structural dependence imposed by previous compression methods, our method creates sub-networks with a lot more flexibility than regular structural pruning methods and also avoids the overhead brought by SliceGPT. The flexibility of our method is reflected in two perspectives. Firstly, our method provides different subsets of the feature maps for different layers. Secondly, our method freely selects the width for each layer without considering architecture dependence. With dramatically increased flexibility, our method outperforms other structural pruning and semi-structural pruning methods given similar pruning ratios. The novel design of the unconstrained pruning space along with strong empirical performance opens new possibilities for structural pruning for LLMs.

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

# A Appendix

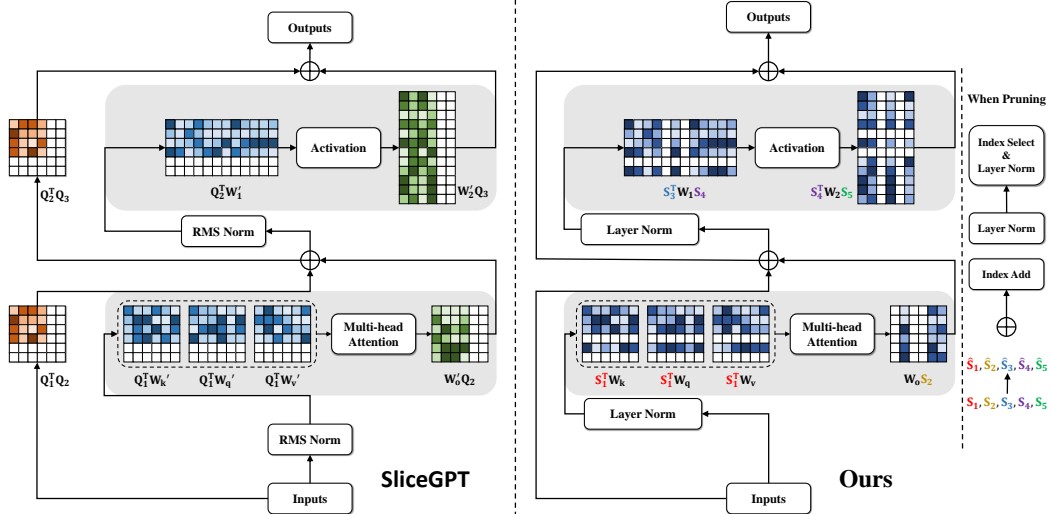

Figure 7: **Left:** SliceGPT inserts $\mathbf{Q}_l^\top \mathbf{Q}_{l+1}$ to the residual connection and brings additional parameters. It also modifies the weights and Layer Norms within the original model. The selection matrix $\mathbf{S}$ is omitted for consistency. **Right:** Our method, DISP-LLM, applies different selection matrices to the input and output dimension of the Attention layer and MLP layer ($\mathbf{S}_1/\mathbf{S}_2$: Attention in/out; $\mathbf{S}_3/\mathbf{S}_4/\mathbf{S}_5$: MLP in/middle/out).

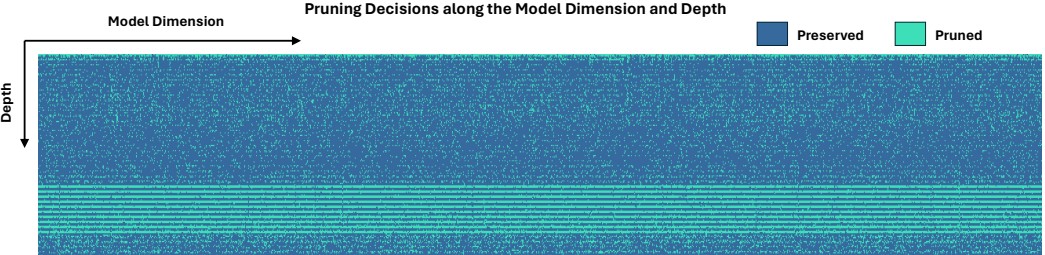

Figure 8: The pruned model architecture along the embedding dimension (model dimension) for the LLaMA-2 13B model when the pruning ratio equals 50%.

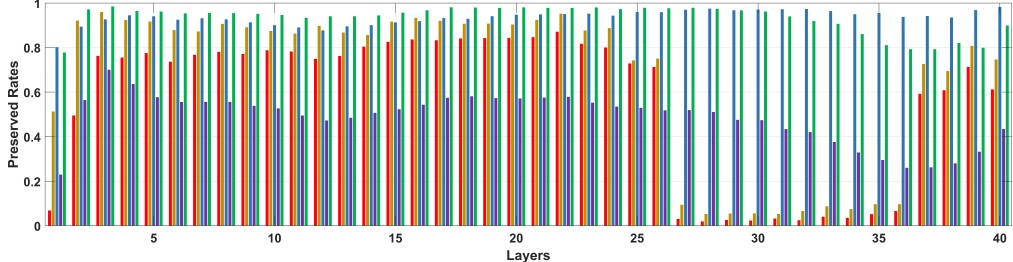

Figure 9: Model width after pruning for the LLaMA-2 13B model when the pruning ratio equals 50%.

## A.1 Proof of Proposition 1

**Proposition 1** (Decreasing feature dimensions for deeper layers). *Let the pseudo-selection matrices in layers $l$ and $l + 1$ be $\mathbf{S}_l$ and $\mathbf{S}_{l+1}$, respectively. The number of nonzero entries in the residual*

---

**Algorithm 2:** Binary ReinMax

---

**Input**: $x$: sigmoid input;
$\tau$: temperature; $c$: constant bias.
**Output**: $\mathbf{x}$: binary vector.
1. $\pi_0 = \text{sigmoid}(x + c)$,
2. $B = \text{sample\_binary}(\pi_0)$,
3. $\pi_1 = \frac{B + \text{sigmoid}((x+c)/\tau)}{2}$,
4. $\pi_1 = \text{sigmoid}(\text{stop\_gradient}(\ln(\pi_1) - (x + c)) + (x + c))$,
5. $\pi_2 = 2\pi_1 - \frac{1}{2}\pi_0$,
6. $\mathbf{x} = \pi_2 - \text{stop\_gradient}(\pi_2) + B$
**Return** $\mathbf{x}$.

---

*adapter satisfies*

$$nnz(\mathbf{S}_l^\top \mathbf{S}_{l+1}) \leqslant \min\{nnz(\mathbf{S}_l), nnz(\mathbf{S}_{l+1})\}.$$

*For compression strategies that remove dependent structures for layer $l + 1$ following $\mathbf{S}_l^\top \mathbf{S}_{l+1}$, this implies that the dimension in layer $l + 1$ is less than or equal to that in layer $l$, with equality holding when the feature indices selected in layer $l + 1$ are contained within those in layer $l$ or vice versa.*

*Proof.* Consider the pseudo-selection matrices $\mathbf{S}_l$ and $\mathbf{S}_{l+1}$, both of size $d \times d$, and both diagonal matrices. The number of nonzero entries in $\mathbf{S}_l$ and $\mathbf{S}_{l+1}$ are given by $nnz(\mathbf{S}_l) = k_l$ and $nnz(\mathbf{S}_{l+1}) = k_{l+1}$, respectively.

The product $\mathbf{S}_l^\top \mathbf{S}_{l+1}$ is also a diagonal matrix of size $d \times d$. Each diagonal entry $(i, i)$ in $\mathbf{S}_l^\top \mathbf{S}_{l+1}$ is the product of the $i$-th diagonal entry of $\mathbf{S}_l$ and the $i$-th diagonal entry of $\mathbf{S}_{l+1}$. For an entry $(i, i)$ to be nonzero, both $\mathbf{S}_l(i, i)$ and $\mathbf{S}_{l+1}(i, i)$ must be nonzero.

Thus, the number of nonzero entries in $\mathbf{S}_l^\top \mathbf{S}_{l+1}$, $nnz(\mathbf{S}_l^\top \mathbf{S}_{l+1})$, is the number of indices $i$ where both $\mathbf{S}_l(i, i)$ and $\mathbf{S}_{l+1}(i, i)$ are nonzero. This count cannot exceed the smaller of the total number of nonzero entries in $\mathbf{S}_l$ and $\mathbf{S}_{l+1}$.

Hence,

$$nnz(\mathbf{S}_l^\top \mathbf{S}_{l+1}) \leqslant \min\{nnz(\mathbf{S}_l), nnz(\mathbf{S}_{l+1})\}.$$

This implies that the effective feature dimension will be smaller or equal to the previous layer. Equality holds if and only if the set of indices corresponding to nonzero entries in $\mathbf{S}_{l+1}$ is a subset of those in $\mathbf{S}_l$, or vice versa. This concludes the proof. $\square$

In Proposition 1, *"remove dependent structures for layer $l + 1$ following $\mathbf{S}_l^\top \mathbf{S}_{l+1}$"* means that the actual selection matrix for layer $l + 1$ becomes $\mathbf{S}'_{l+1} = \mathbf{S}_l^\top \mathbf{S}_{l+1}$, and the structure dependence is cut off by the next residual connection. The pruning for layer $l + 1$ will based on $\mathbf{S}'_{l+1}$ instead of $\mathbf{S}_{l+1}$. Although this setting partially breaks the structural dependence, it has the limitation that the embedding dimensions will be reduced when going deeper.

## A.2 Binary ReinMax

In this section, we provide details for ReinMax when handling binary variables. The ReinMax in our work can be written as shown in Algorithm. 2. We add a constant bias $c$ to $x$ so that we can control binary vectors to have all one value at the beginning when learning the sub-network architecture for DISP-LLMs. Through all experiments, we set $c$ to 3.0 and $\tau$ to 1.0.

## A.3 Details of the Hypernetwork

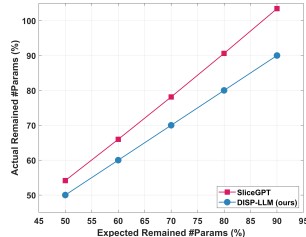

Figure 10: Expected compression rate vs. actual compression rate of our method and Slice-GPT on the LLaMA-7B model.

Table 5: The architecture of hypernetwork.

| Input $z$ |
| --- |
| Bi-GRU(32,64)$\rightarrow$ LayerNorm$\rightarrow$ GeLU |
| Linear$_l$(128, $N_l$)$\rightarrow$Outputs $s_l$, $l = 1, \cdots, L$ |

Table 6: Time costs of our method.

| Model | Time / GPUs |
| --- | --- |
| LLaMA/LLaMA-2 7B | 2.41 Hours / 2 NVIDIA A100 80G |
| LLaMA/LLaMA-2 13B | 8.83 Hours / 4 NVIDIA A100 80G |

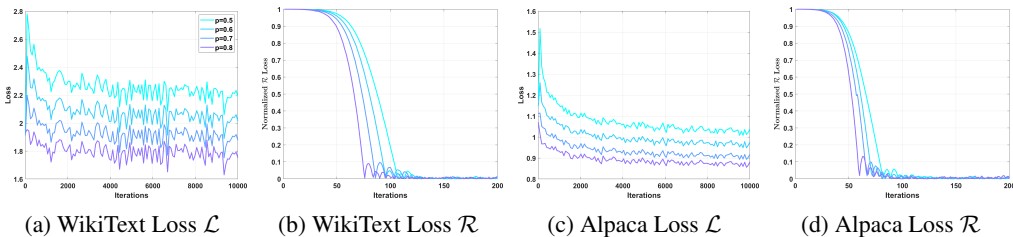

(a) WikiText Loss $\mathcal{L}$     (b) WikiText Loss $\mathcal{R}$     (c) Alpaca Loss $\mathcal{L}$     (d) Alpaca Loss $\mathcal{R}$

Figure 11: The training dynamics when learning the hypernetwork for LLaMA-2 7B model with WikiText and Alpaca datasets.

As we discussed in the paper, the Hypernetwork is composed of linear layers and Bi-GRUs, and now we present the architecture of the HN in Tab. 5. $z$ is initially sampled from a normal distribution, and it is then fixed during training. Outputs $s_l$ are continuous values and it is then fed to ReinMax to produce the binary vector: $\mathbf{s} = \text{ReinMax}(s)$, where $s$ is the collection of $s_l$ from all layers. $N_l$ is the original model width, and it equals the embedding dimension for $\mathbf{S}_1$, $\mathbf{S}_2$, $\mathbf{S}_3$ and $\mathbf{S}_5$.

## A.4 More Implementation Details

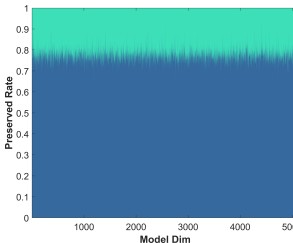

Figure 12: Preserved rates of the LLaMA-2 13B model across different dimensions. The result is accumulated across all the layers.

**Additional Training Details.** During training the hypernetwork, we use AdamW optimizer to optimize it with a constant learning rate $10^{-3}$ and weight decay $0.05$. We train the hypernetwork for different models, we always set the mini-batchsize to $1$ on each GPU. For OPT 6.7B, LLaMA 7B, and LLaMA-2 7B models, we use 2 NVIDIA A100 GPUs, and for LLaMA 13B and LLaMA-2 13B models, we use 4 NVIDIA A100 GPUs. For all the rest models, we use 1 NVIDIA A100 GPU. We set $p = \{0.5, 0.4, 0.3, 0.2, 0.1\}$ when the pruning ratios equals to $\{50\%, 40\%, 30\%, 20\%, 10\%\}$. For the Alpaca dataset [2], we use the 'text' column within the dataset which combines the columns of 'instruction' and 'output'. When training the hypernetwork, we again minimize the language modeling loss on the Alpaca dataset instead of applying the training process of instruction fine-tuning.

**Details of Eq. 6.** The parameter regularization loss function in Eq. 6 is defined as follows:

$$\mathcal{R}(x,y) = \log\left(\frac{\max(x,y)}{\min(x,y)}\right) = \begin{cases} \log\left(\frac{x}{y}\right) & \text{if } x > y, \\ 0 & \text{if } x = y, \\ \log\left(\frac{y}{x}\right) & \text{if } x < y \end{cases} \cdot$$

Since $y$ is fixed, when $x > y$, Eq. 6 will decrease $x$, making it closer to $y$. Conversely, when $x < y$, Eq. 6 will increase $x$, also making it closer to $y$. Thus, the parameter regularization loss always tries to push the current sub-network to achieve the pre-defined parameter budget.

---

[2]https://huggingface.co/datasets/tatsu-lab/alpaca

Table 7: Zero-shot performance of the compressed LLaMA-2 13B model.

| Pruning Ratio | Method | W Update? | WinoGrande acc | HellaSwag acc-norm | ARC-e acc-norm | ARC-c acc-norm | PIQA acc-norm | Avg |
|---|---|---|---|---|---|---|---|---|
| 0% | LLaMA-2 13B | - | 72.22 | 79.39 | 77.48 | 49.23 | 80.47 | 71.76 |
| 30% | SliceGPT [2] | ✗ | 65.11 | 52.69 | 51.77 | 31.23 | 66.10 | 55.16 |
| | K-OBD [34] | ✓ | 64.96 | 64.18 | 56.23 | 36.01 | 74.43 | 59.16 |
| | LLM Surgeon [34] | ✓ | **68.67** | **71.52** | **69.74** | 40.27 | 76.50 | 65.34 |
| | DISP-LLM (Ours) | ✗ | 66.85 | 70.86 | 63.80 | 39.42 | 74.43 | 63.07 |
| | DISP-LLM Alpaca (Ours) | ✗ | 67.32 | 70.04 | 68.98 | **44.28** | **77.31** | **65.59** |
| 40% | K-OBD [34] | ✓ | 60.46 | 55.52 | 49.62 | 32.68 | 70.24 | 53.70 |
| | LLM Surgeon [34] | ✓ | **65.75** | 65.04 | 63.80 | 37.12 | 73.01 | 60.94 |
| | DISP-LLM (Ours) | ✗ | 62.67 | 65.86 | 60.31 | 37.63 | 73.39 | 59.97 |
| | DISP-LLM Alpaca (Ours) | ✗ | 64.25 | **67.52** | **66.79** | **42.75** | **75.30** | **63.32** |
| 50% | K-OBD [34] | ✓ | 57.46 | 48.39 | 46.59 | 30.72 | 66.54 | 49.94 |
| | LLM Surgeon [34] | ✓ | **63.22** | 56.19 | **56.19** | 31.83 | 68.77 | 55.24 |
| | DISP-LLM (Ours) | ✗ | 59.27 | 58.63 | 52.57 | 33.28 | 68.77 | 54.50 |
| | DISP-LLM Alpaca (Ours) | ✗ | 59.59 | **62.39** | 55.72 | **37.54** | **72.20** | **57.49** |

Table 8: Zero-shot performance of the compressed Phi-2 given more pruning rates and settings.

| Pruning Ratio | Method | W Update? | WinoGrande acc | HellaSwag acc-norm | ARC-e acc-norm | ARC-c acc-norm | PIQA acc-norm | Avg |
|---|---|---|---|---|---|---|---|---|
| 0% | Phi-2 | - | 75.61 | 73.86 | 78.24 | 54.01 | 79.11 | 72.17 |
| 20% | SliceGPT [2] | ✗ | **67.80** | 57.76 | 58.00 | 35.32 | 71.87 | 58.15 |
| | +fine-tuning | ✓ | 67.17 | 54.86 | 56.61 | 38.91 | 71.27 | 57.76 |
| | DISP-LLM (Ours) | ✗ | 67.09 | 62.93 | 68.18 | 44.11 | 74.86 | 63.43 |
| 25% | SliceGPT [2] | ✗ | **65.35** | 52.40 | 53.70 | 31.66 | 69.21 | 54.46 |
| | +fine-tuning | ✓ | 65.19 | 52.48 | 52.78 | 35.49 | 69.91 | 55.17 |
| | DISP-LLM (Ours) | ✗ | 65.11 | **59.95** | **65.93** | **43.34** | **74.27** | **61.72** |
| 30% | SliceGPT [2] | ✗ | 63.14 | 47.56 | 53.03 | 30.29 | 65.94 | 51.99 |
| | +fine-tuning | ✓ | 63.54 | 49.72 | 46.38 | 32.68 | 66.16 | 51.70 |
| | DISP-LLM (Ours) | ✗ | **65.19** | **54.43** | **63.59** | **38.48** | **73.34** | **59.00** |

**Ablation Study Settings.** In the ablation study 5.4, we removed the hyperntwork, and we revise Eq. 4:

$$\mathbf{s} = \text{ReinMax}(\Theta),$$

where $\Theta$ now has the same size of $\mathbf{s}$, and the parametrization space becomes much smaller. For the constrained setting used in the ablation study 5.4, we simply let $\mathbf{S}_1 = \mathbf{S}_2 = \mathbf{S}_3 = \mathbf{S}_5$. In section 5.4, we calculate the costs of our method and LLM-Surgeon, and the price comes from the official website of Lambda Cloud[3]. We also list the detailed time costs of our method in Tab. 6. In Fig. 5b, 5f and also in Fig. 11, we normalized the parameter regularization loss $\mathcal{R}$ with its maximum value to make plots more consistent.

**Licenses.** The licenses for various models and datasets are as follows: **LLaMA and LLaMA 2**: Licensed under the LLAMA 2 Community License. **Phi 1.5 and Phi 2**: Licensed under the MIT License. **WikiText dataset**: Licensed under the Creative Commons Attribution-ShareAlike License (CC BY-SA 4.0). **Alpaca dataset**: Licensed under the Creative Commons Attribution-NonCommercial License (CC BY-NC 4.0).

## A.5 Additional Results

**SliceGPT compression rates:** In Fig. 10, we show the expected compression rate and the actual compression rate for our method and SliceGPT given the LLaMA-2 7B model. It can be seen that SliceGPT adds **5% to 13% parameters of the original model** across different pruning rates, which is non-trivial for most LLMs. Notably, SliceGPT with 10% slicing actually adds 3% more parameters to the original model.

**LLaMA-2 13B Results.** In Tab. 7, we show the results of the LLaMA-2 13B model given different pruning rates. From the table, we can see that our method consistently outperforms LLM Surgeon under different pruning rates. The advantage of our method becomes more obvious compared to other methods like K-OBD and SliceGPT.

**Phi-2 Results.** In Tab. 8, we present a more comprehensive comparison of our method compared to SliceGPT. Our method outperforms SliceGPT by 5.28 to 7.01 giving SliceGPT with or without

---

[3]https://lambdalabs.com/service/gpu-cloud#pricing

Table 9: Zero-shot performance of the compressed LLaMA 13B model.

| Pruning Ratio | Method | W Update? | WinoGrande | HellaSwag | ARC-e | ARC-c | PIQA | Avg |
|---|---|---|---|---|---|---|---|---|
| | | | acc | acc-norm | acc-norm | acc-norm | acc-norm | |
| 0% | LLaMA 13B | - | 72.53 | 79.06 | 74.62 | 47.78 | 80.41 | 70.88 |
| 20% | Magnitude | ✗ | 57.54 | 52.90 | 50.13 | 31.14 | 67.57 | 51.86 |
| | +fine-tuning | ✓ | 64.64 | 71.86 | 67.59 | 39.93 | 76.77 | 64.15 |
| | LLM Pruner [30] Channel | ✗ | 58.96 | 49.17 | 49.62 | 31.83 | 66.87 | 51.29 |
| | +fine-tuning | ✓ | 66.38 | 68.89 | 62.08 | 38.99 | 76.55 | 62.58 |
| | LLM Pruner [30] Block | ✗ | 65.11 | 73.41 | 68.35 | 38.40 | 77.15 | 64.48 |
| | +fine-tuning | ✓ | 67.88 | 75.16 | **71.09** | 42.41 | 77.91 | 66.89 |
| | DISP-LLM (Ours) | ✗ | **68.75** | **75.28** | 70.16 | **44.80** | 76.61 | **67.12** |
| | DISP-LLM Alpaca (Ours) | ✗ | 66.54 | 74.80 | 69.73 | 44.71 | **78.07** | 66.77 |
| 50% | DISP-LLM (Ours) | ✗ | **60.85** | 57.81 | 52.51 | 32.51 | 68.44 | 54.42 |
| | DISP-LLM Alpaca (Ours) | ✗ | 59.80 | **58.63** | **56.44** | **34.85** | **71.27** | **56.20** |

Table 10: Average results with 5 different runs. The result is evaluated on WikiText-2.

| Test Performance (PPL), Phi-1.5 Dense: 21.82 | | | | |
|---|---|---|---|---|
| 10% | 20% | 30% | 40% | 50% |
| $18.72 \pm 0.07$ | $20.48 \pm 0.24$ | $22.62 \pm 0.31$ | $25.44 \pm 0.39$ | $32.72 \pm 0.37$ |

Table 11: Throughput of the pruned model.

| Model | Pruning Ratio | Tokens/seconds |
|---|---|---|
| | 0% | 227.99 |
| | 20% | 245.65 |
| LLaMA-2 13B | 30% | 273.60 |
| | 40% | 310.07 |
| | 50% | 342.09 |

fine-tuning on the WikiText dataset. These observations again demonstrate the effectiveness of our method, and our method outperforms methods with recovery fine-tuning in several settings.

**LLaMA-2 3B Results.** In Table 9, we present a comparison of our method against LLM Pruner and magnitude pruning. At a pruning ratio of 20%, our method surpasses the performance of LLM Pruner both with and without fine-tuning. Remarkably, even when the pruning ratio is increased to 50%, our method continues to outperform the LLM Pruner Channel and the Magnitude pruning baselines at a 20% pruning rate. These results further illustrate our method's ability to identify strong sub-networks within the original dense model.

**Architecture of the pruned LLaMA-2 13B.** In Fig. 8 and Fig. 9, we visualize the pruned architecture of the LLaMA-2 13B model pruned with the WikiText dataset. We have similar observations as in section. 5.4. In Fig. 9, we can see that middle to late layers have large pruning rates, especially for the attention layer. In Fig. 8 and Fig. 12, we can also see that the preserved rates for different dimensions are similar, and all embedding dimensions are effectively utilized. More interestingly, similar pruning rates across different dimensions are achieved without adding any regularization or constraints.

**Training loss for LLaMA-2 7B.** In Fig. 11, we further visualize the training dynamics of the LLaMA-2 7B model on WikiText and Alpaca datasets, respectively. The regularization loss with the Alpaca dataset decreases a little bit faster than the WikiText dataset, probably because the loss value on the Alpaca dataset is smaller. From Fig. 11c, we can also see that the language modeling loss continues to decrease when training longer, especially when the pruning ratio is higher.

Lastly, we evaluate our method on the Phi-1.5 model with 5 runs, and we report our result with mean and standard deviation in Tab. 10. We also measure the throughput of our method on the LLaMA-2 13B model, and the result is shown in Tab. 11

### A.6 Additional Analysis

**Impact of $\lambda$ on Phi-1.5 when pruning 50% of parameters.** We show the result in Tab. 12. 'NC' means the loss $\mathcal{R}$ does not converge, and it is much larger than zero, thus it can not prune the model to the target budget. From the table, we can see that the PPL of the model becomes quite stable if it is

Table 12: Impact of $\lambda$ on Phi-1.5

| Test Performance (PPL), Phi-1.5 Dense: 21.82 | | | | | |
|---|---|---|---|---|---|
| $\lambda=1.0$ | $\lambda=2.0$ | $\lambda=4.0$ | $\lambda=6.0$ | $\lambda=8.0$ | $\lambda=10.0$ |
| NC | 36.39 | 33.71 | 32.89 | 33.31 | 33.20 |

larger or equal to 6. If $\lambda$ is not large enough, it takes longer to push the loss $\mathcal{R}$ to reach near 0 values and thus leaves less time for the model to explore different configurations of subnetworks. On the other hand, if $\lambda$ is large enough, the loss $\mathcal{R}$ will reach zero in several hundred iterations and leave enough time to find the desirable subnetwork. Due to this reason, our method is quite stable across larger values of $\lambda$ as shown in the table.

Table 13: PPL vs. pruning ratio trade-off for the Phi-2 model.

| Test Performance (PPL), Phi-2 Dense: 10.98 | | | | |
|---|---|---|---|---|
| 10% | 20% | 30% | 40% | 50% |
| 10.22 | 10.94 | 14.46 | 16.02 | 20.05 |

Table 14: Zero-shot task performance vs pruning ratio trade-off for the LLaMA-2 7B model with the WikiText dataset

| Avg task acc, LLaMA-2 7B | | | | |
|---|---|---|---|---|
| 0% | 20% | 30% | 40% | 50% |
| 68.99 | 62.54 | 58.10 | 52.63 | 46.72 |

Table 15: Zero-shot performance of the compressed LLaMA-2 7B model with LoRA fine-tuning.

| Pruning Ratio | Method | W Update? | WinoGrande | HellaSwag | ARC-e | ARC-c | PIQA | Avg |
|---|---|---|---|---|---|---|---|---|
| | | | acc | acc-norm | acc-norm | acc-norm | acc-norm | |
| 0% | LLaMA 7B | - | 69.85 | 76.21 | 72.81 | 44.71 | 79.16 | 68.55 |
| 50% | DISP-LLM Alpaca | ✗ | 56.20 | 49.35 | 51.14 | 30.20 | 68.34 | 51.05 |
| | +LoRA ft | ✓ | 56.83 | 53.74 | 57.49 | 32.42 | 70.78 | 54.25 |

**More results on pruning ratio vs. performance trade-offs.** We provide the trade-off between the pruning ratio and performance for the Phi-2 and LLaMA-2 7B model below in Tab. 13 and Tab. 14.

**LoRA fine-tuning [18] of the compressed LLaMA-2 7B model.** We follow similar settings of the SliceGPT and the model is fine-tuned on Alpaca. The result is shown in Table. 15. We can see that our method can be further boosted by using parameter-efficient fine-tuning techniques. Since the performance without fine-tuning is already good enough, we prefer not to involve this additional process in our method to save time and computational costs.

## A.7    Generation Samples

We show the generated text given DISP-LLM and SliceGPT in Tab. 16. The examples are obtained based on removing 20% of the model weights with DISP-LLM and 20% slicing of SliceGPT ( 10% compression rate). Both models are compressed from LLaMA-2 7B and they are not finetuned. From these two examples, we can see that SliceGPT only generates a small part of meaningful content and then starts repeating itself. On the other hand, our method tends to generate more relevant content, and the self-repeating behavior is much less obvious. These observations comply with the performance gap between DISP-LLM and SliceGPT.

## A.8    Limitations

Our method explores how to break the structural dependency of pruning for LLMs. Although our method achieves competitive performance, there are still some limitations that are not solved in our current version. The throughput improvements of our method are not consistent across different models, which is probably because the current implementation of index add or index select operations with PyTorch is not efficient enough. An alternative implementation of our method is to select weight matrices instead of feature maps. In this approach, we perform matrix multiplication with the full feature map and fill the weight matrices with zeros. If we have a custom matrix multiplication implementation that ignores zero rows or columns, it may achieve further speed-up. However, this approach primarily focuses on the engineering perspective and is beyond the scope of this work. We leave the exploration of this alternative implementation to future research.

## A.9    Border Impact

Our dimension-independent structural pruning method significantly reduces the computational and memory requirements of LLMs, enabling their deployment on resource-limited devices and lowering energy consumption, thus contributing to environmental sustainability. By facilitating the use of efficient LLMs on affordable hardware, our approach democratizes access to advanced AI technologies. However, it is crucial to address potential ethical concerns when reducing the size of LLMs, ensuring privacy and fairness in AI applications with compressed LLMs. Overall, our method fosters a more accessible and sustainable use of AI.

Table 16: Generation samples of LLaMA-2 7B with 20% compression rate

| | |
|---|---|
| **SliceGPT** | **AI is** a very important part of the game. The game is played in a 3D environment, the player controls a character that can move freely around the environment. The player can interact with the environment by picking up items, using them to solve puzzles. The player can also interact with other characters in the game. The player can also interact with the environment by picking up items and using them to solve puzzles. The player can also interact with other characters in the game. The player can also interact with the environment by picking up items and using them to solve puzzles. The player can also interact with other characters in the game. The player can also interact with the environment by picking up items and using them to solve puzzles. The player can also interact with other characters in the game. The player can also interact with the environment by picking up items and using them to solve puzzles. The player can also interact with

**The room-temperature superconductor is** a material that exhibits superconductivity at room temperature.(100 ℃). The first room temperature superconductor was discovered in 1970 by Peter C. Edwards and his colleagues at the University of Michigan. The material was a mixture of bismuth and tin, and was named " bismuth- tin." The discovery of bismuth tin was followed by the discovery of other room temperature superconductors, including bismuth tin, bismuth tin, bismuth tin, bismuth tin, bismuth tin, bismuth tin, bismuth tin, bismuth tin, bismuth tin, bismuth tin, bismuth tin, bismuth tin, bismuth tin, bismuth tin, bismuth tin, bismuth tin, bismuth tin, bismuth tin, bismuth tin, bismuth tin, bismuth tin, bismuth tin, bismuth tin, bismuth tin, bismuth tin, bismuth tin, bismuth tin, bismuth tin, bismuth tin, bismuth tin, bismuth tin, bismuth tin, bismuth tin, bismuth tin, bismuth tin, bismuth tin, bismuth tin, bismuth tin, bismuth tin, bismuth tin, bismuth tin, bismuth tin, bismuth tin, bismuth tin, bismuth tin, bismuth tin, bismuth tin, bismuth tin, bismuth tin, bismuth tin, bismuth tin, bismuth tin, bismuth tin, bismuth tin, bismuth tin, bismuth tin, bismuth tin, bismuth tin |
| **DISP-LLM** | **AI is** a branch of computer science that seeks to create intelligent machines that can think and learn like humans. The field was developed in the 1950s and 1960s by researchers who focused on making machines that could learn and solve problems like people do. The field was commercially successful in the 1980s and 1990s with the development of the first generation of intelligent agents. The term " artificial intelligence " was coined by John McCarthy in 1956. He was inspired by the idea of creating a machine that could learn and solve problems like humans do.

**The room-temperature superconductor is** a class of superconductors that exhibit zero resistance at room temperature. ## History The room-temperature superconductor was discovered in 1986 by the Japanese scientist K. Masamichi Aoki and his colleagues at the University of Tokyo. The discovery was made possible by the use of a new technique called " zero temperature transport measurement " ( ZTM ), which allowed them to measure the resistance of the superconductor at temperatures as low as 0.05 K. The discovery was made possible by the use of a new technique called " zero temperature transport measurement " ( ZTM ), which allowed them to measure the resistance of the superconductor at temperatures as low as 0.05 K. ## Discovery The discovery of the room-temperature superconductor was made possible by the use of a new technique called " zero temperature transport measurement " ( ZTM ), |

