# OpenReview forum: "DISP-LLM: Dimension-Independent Structural Pruning for Large Language Models"
_NeurIPS.cc/2024/Conference — NeurIPS 2024 poster_

### Official Review · Reviewer_BGu5 · 2024-07-11

**Soundness:** 4
**Presentation:** 4
**Contribution:** 3
**Rating:** 5
**Confidence:** 4

**Summary:**

This paper concerns about the dimension dependence of hidden states across different layers. It have been argued that previous studies with dimension dependence would be inefficient due to additional parameters or architectural constraints. This work breaks the dimension dependence by introducing indexing operations to previous methods. Although simple, the proposed method is shown to be more effective than baselines due to permitted asymmetry across layers.

**Strengths:**

1. It is for the first time highlighted that the dimension dependence may affect the performance of pruning.
2. The proposed method to alleviate the dimension dependence is rather simple and effective.

**Weaknesses:**

1. Although the proposed method allows asymmetric dimensions in module weights across layers, it also requires the dimensions of hidden states from different layers maintain the same as that of the non-pruned model. This would restrict the use in scenarios where the dimension of the output hidden states is critically important (e.g., embedding for retrieval).
2. The introduced indexing operation could be very time-consuming. Unexpectedly, the latency introduced by the indexing may outweight the latency reduced by pruning. And it is not discussed whether the pruned model is faster than the non-pruned one. I would recommend including such experiments in proper positions.

**Questions:**

N/A

**Limitations:**

Yes

---

> ### Author Rebuttal · Authors · 2024-08-06
>
> We want to thank the Reviewer BGu5 for your insightful and constructive comments. We will address each of your questions below. We use W1 to refer to weakness 1.
>
> **W1. Although the proposed method allows asymmetric dimensions in module weights across layers, it also requires the dimensions of hidden states from different layers maintain the same as that of the non-pruned model. This would restrict the use in scenarios where the dimension of the output hidden states is critically important (e.g., embedding for retrieval).**
>
> In our method, the model dimension (the dimensions of hidden states) remains unchanged after pruning, which is also the prerequisite for dimension-independent pruning. Consequently, our model can serve as a drop-in replacement for the original dense model in any downstream tasks, such as retrieval, as the feature map dimension will stay the same as the original model.
>
> If this does not address your concern about the dimensions of hidden states and their impact on scenarios like embedding for retrieval, could you please provide more details or clarify your point further? We want to ensure we fully understand your concern and address it appropriately.
>
> **W2. The introduced indexing operation could be very time-consuming. Unexpectedly, the latency introduced by the indexing may outweight the latency reduced by pruning. And it is not discussed whether the pruned model is faster than the non-pruned one. I would recommend including such experiments in proper positions.**
>
> We present the acceleration results in Figure 5 (g) and Table 10 for the LLaMA-2 13B model, demonstrating that our model is faster than the non-pruned model. While we observe some speed improvements with our current model, we believe there is significant room for further enhancement, as the current PyTorch implementation of indexing operations is not efficient enough. Additionally, the theoretical FLOPs of indexing operations are very low. This is also discussed in the Limitation section of our paper. We believe the speed of our method can be further improved by incorporating a custom implementation of the matrix multiplication algorithm, where pruned columns and rows are skipped instead of using indexing operations. However, this approach primarily focuses on the engineering perspective and is beyond the scope of this work.

---

> > ### Comment · Area_Chair_3Hzz · 2024-08-13
> >
> > Dear Reviewer BGu5,
> >
> > Thanks again for helping review this paper! Since we are approaching the end of the author-reviewer discussion period, would you please check this author response regarding your concerns? We really appreciate it!
> >
> > Best, AC

---

### Official Review · Reviewer_AWnP · 2024-07-12

**Soundness:** 3
**Presentation:** 2
**Contribution:** 3
**Rating:** 5
**Confidence:** 4

**Summary:**

DISP-LLM introduces dimension-independent structural pruning for LLMs, breaking structural dependencies to allow flexible pruning across layers and dimensions. It uses a hypernetwork for pruning decisions and claims superior performance over existing methods on various LLMs without weight updates or additional parameters.

**Strengths:**

- Novel approach to structural pruning in LLMs
- Comprehensive evaluation across multiple LLM architectures
- Theoretical foundation provided
- Achieves good performance without weight updates or extra parameters

**Weaknesses:**

- Limited novelty beyond combining existing concepts
- Inconsistent performance gains across models/tasks
- Insufficient ablation studies and hyperparameter sensitivity analysis
- Lack of detailed computational efficiency comparisons
- Inadequate error analysis and failure case exploration
- Overstated claims based on limited experiments
- Insufficient discussion of pruning ratio vs. performance trade-offs

**Questions:**

- How sensitive is DISP-LLM to hypernetwork architecture and hyperparameters?
- Can you provide detailed computational cost comparisons with baselines?
- How does it perform on fine-tuning tasks?
- Have you explored progressive pruning approaches?

**Limitations:**

Yes

---

> ### Author Rebuttal · Authors · 2024-08-06
>
> We want to thank the Reviewer AWnP for your insightful and constructive comments. We will address each of your questions below. We use W1 to refer to weakness 1 and Q1 to refer to question 1.
>
> **W1. Limited novelty beyond combining existing concepts**
>
> Our method is novel because it introduces a new pruning configuration by breaking structural dependencies, significantly expanding the potential search space for an optimal sub-network which is not presented in existing works. This increased search space makes most methods that rely on static metrics (such as weight norms) ineffective. Additionally, reinforcement learning or evolutionary search algorithms struggle in this context due to the difficulty of managing fine-grained search requirements, such as deciding whether to prune a single dimension. To address this, we employ gradient-based search as a highly efficient tool. From this perspective, our method is novel because we demonstrate that increasing the capacity (using latent hypernetworks) for gradient-based search is a highly effective approach for handling large and complex search spaces, such as DISP-LLM.
>
> **W2. Inconsistent performance gains across models/tasks**
>
> We believe it is reasonable that the performance gains for different models are different, as they are pre-trained with different data and computational costs. Models trained with higher costs are naturally harder to prune. For instance, consider the perplexity of OPT 6.7B and LLaMA-2 7B across different compression rates. It is possible to achieve lower perplexity than the dense model with OPT 6.7B (9.89 vs. 10.86), whereas it is impossible for LLaMA-2 7B (6.10 vs. 5.12), both of which are at a 20% compression rate. LLaMA-2 7B is often regarded as a better pre-trained model than OPT 6.7B, and it is also more challenging to prune. Thus, the performance of our method is naturally different for these two models.
>
> We also think that performance gains across different tasks should vary since different tasks have different emphases. A single, small dataset like Wikitext or Alpaca cannot cover all perspectives, resulting in varied performance gains across different tasks.
>
> If the above explanation does not address your concern about inconsistent performance gains across models/tasks, could you please provide more details? We would be happy to address any specific aspects you are referring to.
>
> **W3 & Q1: Insufficient ablation studies and sensitivity analysis for hypernetwork architecture and hyperparameters**
>
> Please see the first part of the general response.
>
> **W4 & Q2: Lack of detailed detailed computational cost comparisons.**
>
> We listed the detailed computational cost comparison below:
>
> **Computational Costs comparisons**
>
> Costs | DISP-LLM | LLM Suregon
> ---|---|---
> LLaMA-2 7B | 2h25m / 2 NVIDIA A100 80G | 16h58m / 4xH100 80 GB
> LLaMA-2 13B | 8h50m / 4 NVIDIA A100 80G | 1d6h5m / 8xH100 80 GB
>
>
> The computational cost of our method is also listed in Table 5 in the Appendix. Compared to LLM Suregon, our method is 14.76 x to 27.39 x cheaper in terms of US dollars.
>
> **W5. Inadequate error analysis and failure case exploration**
>
> Please see the second part of the general response.
>
> **W6. Overstated claims based on limited experiments**
>
> We believe our experimental results are accurate and not overstated. Firstly, our method surpasses semi-structural pruning methods in terms of perplexity at the same compression rate. To the best of our knowledge, this is the first time that a structural pruning method outperforms semi-structural pruning methods in perplexity. Additionally, as mentioned in the contribution section, our method outperforms other baseline methods, with or without fine-tuning, without requiring updated model weights. This claim is also well reflected in the experimental section.
>
> If there are specific claims or aspects of our experiments you find overstated or insufficiently supported, please let us know. We would appreciate more details so we can address your concerns thoroughly.
>
> **W7. Insufficient discussion of pruning ratio vs. performance trade-offs**
>
> Please see the third section of the general response.
>
> **Q3. How does it perform on fine-tuning tasks?**
>
> We further evaluate our method on the Vicuna 7B model, which is a model trained by fine-tuning the LLaMA 7B model. We list the results below:
>
> **Average zero-shot task performance with the Vicuna 7B model**
>
> Methods | Pruning Ratio | W Update? | WinoGrande | HellaSwag | ARC-e | ARC-c | PIQA | Avg
> ---|---|---|---|---|---|---|---|---
> Vicuna 7B | 0% | - | 67.40 | 70.64 | 65.11 | 41.21 | 77.75 | 64.42
> LLM-Pruner | 50% | No | 50.28 | 34.86 | 33.29 | 27.3 | 59.79 | 41.10
> w finetuning | 50% | Yes | 54.54 | 46.79 | 48.15 | 29.78 | 69.84 | 49.82
> DISP-LLM Alpaca (Ours) | 50% | No | 55.72 | 46.57 | 52.31 | 30.38 | 66.70 | 50.34
>
> The result above is consistent with pre-trained models, and our method can be seamlessly applied to models fine-tuning on different tasks.
>
> **Q4. Have you explored progressive pruning approaches?**
>
> For now, we have not explored the progressive pruning approach. Progressive pruning typically involves updating model weights through either full fine-tuning or LoRA fine-tuning, which would significantly increase the computational costs of our method. Given the current strong performance of our method, we aim to maintain its low cost without introducing an additional weight training process.

---

> > ### Comment · Area_Chair_3Hzz · 2024-08-13
> >
> > Dear Reviewer AWnP,
> >
> > Thanks again for helping review this paper! Since we are approaching the end of the author-reviewer discussion period, would you please check this author response regarding your concerns? We really appreciate it!
> >
> > Best, AC

---

### Official Review · Reviewer_6CTn · 2024-07-13

**Soundness:** 3
**Presentation:** 3
**Contribution:** 3
**Rating:** 6
**Confidence:** 3

**Summary:**

This paper proposes a novel structural pruning method for LLMs, primarily based on SliceGPT. It characterised in (i) removing structural dependence by facilitating each block to possess varying widths along its input and output dimensions (ii) no need for introducing addition paramaters like sliceGPT.  The empirical studies show that it demonstrates superior performance compared to state-of-the-art structural pruning techniques for LLMs across a range of models, including OPT, LLaMA, LLaMA-2, Phi-1.5, and Phi-2.

**Strengths:**

1. This paper is easy-to-read and topic is very important to efficient Machine Learning.
2. The proposed method balances the task performance and efficient, which are both beneficial to its practical and wide application.
3. The empirical results show its promising in model compressing across four models.

**Weaknesses:**

1.  More comprehensive literature review about model compressing, like sparse mechanism.
2. Any case study for qualitative analysis to the pruned model? It is possible to conduct some interpretability analysis showing where the difference between sliceGPT and the proposed method along with a specific input example.

**Questions:**

1.  When training with the pruning objectives all the layers are fine-tuned or only selected layers?
2. Any insights from Figure 6, different model width as layer changes. If the observations aligned with existing research results?

**Limitations:**

Pay attention to the generated sequence after model prune because the prediction changes are unpredictable.

---

> ### Author Rebuttal · Authors · 2024-08-06
>
> We want to thank the Reviewer 6CTn for your insightful and constructive comments. We will address each of your questions below. We use W1 to refer to weakness 1 and Q1 to refer to question 1.
>
> **W1. More comprehensive literature review about model compressing, like sparse mechanism.**
>
> Thanks for pointing this out. We will add more related works regarding sparse mechanisms, like ‘The Lottery Ticket Hypothesis’ and other works related to sparse mechanisms and model compression.
>
> **W2. Any case study for qualitative analysis to the pruned model? It is possible to conduct some interpretability analysis showing where the difference between sliceGPT and the proposed method along with a specific input example.**
>
> We provided two examples in Table 1 of the rebuttal pdf file. The examples are obtained based on removing 20% of the model weights with DISP-LLM and 20% slicing of SliceGPT (~10% compression rate). Both models are compressed from LLaMA-2 7B and they are not finetuned. From these two examples, we can see that SliceGPT only generates a small part of meaningful content and it then starts to repeat itself. On the other hand, our method tends to generate more relevant content, and the self-repeating behavior is much less obvious. These observations comply with the performance gap between DISP-LLM and SliceGPT.
>
> **Q1. When training with the pruning objectives all the layers are fine-tuned or only selected layers?**
>
> We do not fine-tune model weights. Throughout the whole training process, only the hypernetwork is trained and thus the total trainable parameters of our method are quite small which is usually around 2-3% of the original model weights. The hypernetwork is only responsible for controlling the selection matrices of different layers as shown in Figure 2.
>
> **Q2. Any insights from Figure 6, different model width as layer changes. If the observations aligned with existing research results?**
>
> From Figure 6, we can observe that the middle layers of the model are better kept. This may comply with some interpretability works. In [1], they argue that the middle layer of the model is likely to contain interesting, abstract features.
>
> On the other hand, our method does not retain as much of the first and last layers as LLM Surgeon does. This is probably the result of removing structural dependency so that each layer has a larger flexibility.
>
> We believe that future works are needed to fully understand the importance of different layers under our dimensional independent setting.
>
> [1] Templeton, A. (2024). Scaling monosemanticity: Extracting interpretable features from Claude 3 sonnet. Anthropic.

---

> > ### Comment · Reviewer_6CTn · 2024-08-10
> >
> > Thank you for your clarification and the case study you provided—it was very helpful!
> > I find the observations in Figure 6 particularly interesting, especially how both the earlier and later layers are changing significantly. A similar study referenced in [1] also presents insights on layer-wise monosemanticity, which I hope might offer some additional inspiration.
> >
> > Encourage or Inhibit Monosemanticity? Revisit Monosemanticity from a Feature Decorrelation Perspective

---

> > > ### Author Response · Authors · 2024-08-11
> > >
> > > We would like to thank Reviewer 6CTn for providing the reference, which is an excellent addition to explaining the pruned structure. It is possible that the retained capacity is related to the monosemanticity of the corresponding layer. We will add the related discussion to our final version to provide further insights regarding the pruned structure.

---

### Official Review · Reviewer_FhgQ · 2024-07-13

**Soundness:** 3
**Presentation:** 3
**Contribution:** 3
**Rating:** 6
**Confidence:** 3

**Summary:**

Structural pruning is a method to prune the weights of large language models (LLMs) while keeping their original performance as much as possible it can. However, structural pruning has limitations caused by depending on the structure, like the residual connection of LLMs. In this work, the authors propose a new pruning method, dimension-independent structural pruning for LLMs (DISP-LLM), that can ignore structural dependence in pruning. This characteristic of DISP-LLM makes it possible to prune their weights more flexibly and provides new pruning ways using different feature maps between blocks and varying pruning dimensions of the input and output of the block. However, just pruning the dimensions and ignoring residual connections causes inconsistency between the input and output of a block. To solve this problem, the authors apply transformation matrices to each residual connection to absorb these discrepancies. Furthermore, pruned indices are optimized through training. Experimental results on the language modeling dataset WikiText-2 and the zero-shot task datasets PIQA, WinoGrande, HellaSwag, ARC-e, and ARC-c show that DISP-LLM tends to outperform conventional approaches in pruning LLMs, OPT, LLaMA, LLaMA-2, Phi-1.5, and Phi-2.

**Strengths:**

- The proposed dimension-independent structural pruning for LLMs (DISP-LLM) can drastically prune dimensions due to its flexibility.
- Adding transformation matrices to each residual connection to maintain input and output consistency in each transformer is novel
- In DISP-LLM, pruned dimensions can be optimized through training.
- Experiments are comprehensive and cover various kinds of LLMs.
- DISP-LLM can achieve comparable or better perplexities to conventional pruning approaches that require updating the original weights by training.
- In many cases, DISP-LLM outperforms conventional pruning approaches in zero-shot tasks even though some of them update the original weights by training.
- The authors report the actual costs for pruning models and show the cost efficiency of DISP-LLM.

**Weaknesses:**

- Even though DISP-LLM does not update the original weights of LLMs, training is required.
- DISP-LLM requires hyper-tuning for $\lambda$.

**Questions:**

- How did you decide to set $\lambda$ to 6?
- How are the performances sensible to the changes of $\lambda$?
- As you state in the paper, DISP-LLM is as efficient as LoRA. If so, can applying LoRA weights only to the kept dimensions be a baseline?

**Limitations:**

DISP-LLM requires training and hyper-parameter tuning for $\lambda$ that controls mixing ratios of losses.

---

> ### Author Rebuttal · Authors · 2024-08-06
>
> We want to thank the Reviewer FhgQ for your insightful and constructive comments. We will address each of your questions below. We use W1 to refer to weakness 1 and Q1 to refer to question 1.
>
> **W1. Even though DISP-LLM does not update the original weights of LLMs, training is required.**
>
> In fact, the training of our method is very lightweight, as we pointed out in the response to **Q3**. Moreover, with a small training costs, our method largely outperforms previous pruning methods which suggests that our method is very cost-effective. In addition, freezing model weights give us potential opportunities to further reduce the costs. For example, we can quantize the original model weights to lower bits to further reduce its GPU memory costs. After pruning, we can restore the model to the original precision. If a method needs to train model weights during pruning, such as LLM Surgeon, it must implement the complex pipeline of quantization-aware training. In contrast, our method does not require this because we do not update the weights of the original model.
>
> **W2 & Q1 & Q2. How to decide $\lambda$ to 6 and the performance sensitivity to $\lambda$.**
>
> Please refer to the first section of the general response. Overall speaking, the hyperparameter selection of our method is not complex.
>
> **Q3. As you state in the paper, DISP-LLM is as efficient as LoRA. If so, can applying LoRA weights only to the kept dimensions be a baseline?**
>
> Indeed, the trainable parameters of our method are around 2-3% of the original model weights which is at a similar scale to LoRA. Though the training cost is similar, applying LoRA does not remove any model weights or reduce the model size. Thus, it may not be a useful baseline. If we apply LoRA weights to the pruned model, then it becomes recovery finetuning, which is definitely doable and we present some results here:
>
> **LoRA finetuning of the compressed LLaMA-2 7B model.**
>
> Methods | Pruning Ratio | W Update? | WinoGrande | HellaSwag | ARC-e | ARC-c | PIQA | Avg
> ---|---|---|---|---|---|---|---|---
> DISP-LLM Alpaca | 50% | No | 56.20 | 49.35 | 51.14 | 30.20 | 68.34 | 51.05
> w LoRA ft | 50% | Yes | 56.83 | 53.74 | 57.49 | 32.42 | 70.78 | 54.25
>
> We follow similar settings of the SliceGPT and the model is fine-tuned on Alpaca. We can see that our method can be further boosted by using parameter-efficient finetuning techniques. Since the performance without fine-tuning is already good enough, we prefer not to involve this additional process in our method to save time and computational costs.

---

> > ### Comment · Reviewer_FhgQ · 2024-08-12
> > **Regarding LoRA**
> >
> > Thank you for answering my questions! The response has resolved some of my concerns. Regarding the use of LoRA, one of the frequently used approaches is QLoRA [1], which uses LoRA on quantized frozen weights. Even though QLoRA does not prune the model weight, it accomplishes producing lightweight models with fine-tuned information. Therefore, considering the quantization in the comparison is along with the realistic situation.
> > As seen in the general response, the author seems to notice the necessity of investigating the use of quantization in the proposed method. Currently, this direction has not been decided in the paper. However, if you can clarify the difference between the proposed method, quantization, and QLoRA in terms of their benefits and still claim the superiority of your approach to other methods, I can raise my score.
> >
> > [1] Dettmers, Tim, et al. "Qlora: Efficient finetuning of quantized llms." Advances in Neural Information Processing Systems 36 (2024).

---

> ### Author Response · Authors · 2024-08-12
>
> We want to thank Reivewer FhgQ for giving us the chance for further clarification. We list the comparison of our method, quantization, and QLoRA in the table below:
>
>  Methods | W Update? | Reduce Weight Precision | Reduce #Parameters | Goal
> ---|---|---|---|---
>  DISP-LLM | No | No | Yes | Compression
>  Quantization | Depends | Yes | No | Compression
>  QLoRA | Yes | Yes | No | Fine-tuning
>
> As a pruning method, DISP-LLM is orthogonal to Quantization or QLoRA methods as shown in the table, meaning it can be combined with quantization to further reduce model size. Firstly, we can combine our method and Quantization by applying DISP-LLM on top of a quantized model to reduce the training costs as stated in the response to **W1**. Additionally, once a model has been compressed using DISP-LLM, QLoRA can be employed to fine-tune the model with minimal resources. Besides the aforementioned contents, we believe that the search space in the dimensional independent setting offers new insights for designing a compact large language model that is not captured by QLoRA or quantization.

---

### Author Rebuttal · Authors · 2024-08-06

General Response to Reviewers:

We would like to thank all reviewers for their insightful comments. We are greatly encouraged to see that all of you hold a positive evaluation of our work. We will address some common comments in the general response. We also put part of the individual response here due to the space limitation. All relevant results within the Rebuttal will be added to the final version of our paper.

**1. How to select $\lambda$ and how  $\lambda$ affects the performance and insufficient ablation study. (Reviewer FhgQ and Reviewer AWnP):**

We present the perplexity of Phi-1.5 for various values of $\lambda$, as shown in the results below:

**The impact of Different  $\lambda$ on the Phi-1.5 model**

  $\lambda$ | 1 | 2 | 4 | 6 | 8 | 10
---|---|---|---|---|---|---
PPL | NC | 36.39 | 33.71 | 32.89 | 33.31 | 33.2

‘NC’ means the loss R does not converge, and it is much larger than zero, thus it can not prune the model to the target budget. From the table, we can see that the PPL of the model becomes quite stable if it is larger or equal to 6. If  $\lambda$ is not large enough, it takes longer to push the loss R to reach near 0 values and thus leaves less time for the model to explore different configurations of subnetworks. On the other hand, if  $\lambda$ is large enough, the loss R will reach zero in several hundred iterations and leave enough time to find the desirable subnetwork. Due to this reason, our method is quite stable across larger
values of  $\lambda$ as shown in the table.

We add the ablation studies regarding the hypernetwork architecture below:

**The impact of Hypernetwork architecture on the Phi-1.5 model. Performance is measured by perplexity**

Settings/Compression Rate | 10% | 20% | 30% | 40% | 50%
---|---|---|---|---|---
w/o Linear layers and GRU | 20.37 | 22.30 | 28.66 | 34.33 | 47.29
w/o Bi-GRU | 19.90 | 21.65| 26.11 | 30.88 | 37.43
Full Hypernet | 18.75 | 20.23 | 22.81 | 25.49 | 32.89

For the 'w/o Bi-GRU' setting, we simply remove GRU and add a fixed input (initialized in the same way as $z$ see Appendix A.3 for more details) for each Linear layer. The results indicate that both GRU and Linear layers affect the final performance. A reasonable explanation is that Linear layers connect different dimensions of the model, accelerating learning, while GRU layers capture inter-layer relationships, further reducing the difficulty of learning subnetwork structures. And thus both of them contribute to the final performance.

**2. Inadequate error analysis and failure case exploration (Reviewer AWnP)**

There is one failure case that we want to mention in the response which is shown in the first part of the general response. In the table, we can see that if  $\lambda$= 1 ('NC' not converge for R loss), our method fails to push R loss to near 0 values (<0.01), and thus it can not prune the model to the given budget. This suggests that we should avoid selecting a too small value of  $\lambda$. For the rest of the $\lambda$ values, our method is robust.



**3. Insufficient discussion of pruning ratio vs. performance trade-offs (Reviewer AWnP)**

In the experimental section, we already provided the pruning ratio vs. perplexity for the Phi-1.5 model at 5 different pruning rates as shown in the blue line (DISP-LLM w HN) in Figure 5. (c). Furthermore, the pruning ratio vs. perplexity for OPT and LLaMA-2 models are given in Table 1. We further provide the trade-off between the pruning ratio and performance for the Phi-2 and LLaMA-2 7B model below:

**PPL vs compression rate trade-off for the Phi-2 model**

Compression Rate | 0%  | 10% | 20% | 30% | 40% | 50%
---|---|---|---|---|---|---
PPL | 10.98 | 10.22 | 10.94 | 14.46 | 16.02 | 20.05

**Zero-shot task performance vs compression rate trade-off for the LLaMA-2 7B model with the WikiText dataset**

Compression Rate  | 0% |  20% | 30% | 40% | 50%
 ---|---|---|---|---|---
 Avg acc | 68.99 | 62.54 | 58.10 | 52.63 | 48.74

We believe with these results, the readers should have a better understanding of the trade-off between performance and compression rates for our method.

---

### Decision · Program_Chairs · 2024-09-25

**Decision:**

Accept (poster)

**Comment:**

This paper introduces an innovative method of structural pruning that does not rely on structural dependence among embedding dimensions. The dimension independency nature of this method can significantly increase the flexibility of structural pruning. Experimental results demonstrate its effectives on various open-source LLMs by improving upon state-of-the-art methods. Overall, this is a solid work, and its contribution is appreciated by reviewers. The rebuttal addressed the reviewer concern by clarifying that this approach requires lightweight training and hyper-parameter selection. The consensus among reviewers emerged through rebuttal and discussion. Authors are encouraged to include these additional results and discussions into the final version.